New mammalian and avian records from the late Eocene La Meseta and Submeseta formations of Seymour Island, Antarctica

Davis Sarah N. sdavis6@utexas.edu 1
Torres Christopher R. 2
Musser Grace M. 1
Proffitt James V. 3
Crouch Nicholas M.A. 1
Lundelius Ernest L. 1
Lamanna Matthew C. 4
Clarke Julia A. 1
1 Department of Geological Sciences, Jackson School of Geosciences, The University of Texas at Austin , Austin , TX , United States of America
2 Department of Integrated Biology, The University of Texas at Austin , Austin , TX , United States of America
3 School of Medicine, The University of Missouri , Columbia , MO , United States of America
4 Section of Vertebrate Paleontology, Carnegie Museum of Natural History , Pittsburgh , PA , United States of America
Farke Andrew
Electronic publication date: 2020 Jan 9
Publication date: 2020
Volume: 8
Electronic Location ID: e8268
Received 2019 Jul 15; Accepted 2019 Nov 22
Copyright: ©2020 Davis et al.
Copyright year: 2020
Copyright holder: Davis et al.
License: This is an open access article distributed under the terms of the Creative Commons Attribution License, which permits unrestricted use, distribution, reproduction and adaptation in any medium and for any purpose provided that it is properly attributed. For attribution, the original author(s), title, publication source (PeerJ) and either DOI or URL of the article must be cited.
License URL: https://creativecommons.org/licenses/by/4.0/

Keywords: Xenarthra, Gruiformes, Sphenisciformes, Eocene, Seymour Island, Antarctica, La Meseta, Submeseta, Biogeography, Mammal, Bird

Funding: National Science Foundation Office of Polar Programs ANT-1141820 ANT-1142129 ANT-1142102 ANT-1142052 This project was supported by the National Science Foundation Office of Polar Programs grants to the AP3 team: NSF OPP ANT-1141820, ANT-1142129, ANT-1142102, and ANT-1142052. The funders had no role in study design, data collection and analysis, decision to publish, or preparation of the manuscript.

==============================
The middle–late Eocene of Antarctica was characterized by dramatic change as the continent became isolated from the other southern landmasses and the Antarctic Circumpolar Current formed. These events were crucial to the formation of the permanent Antarctic ice cap, affecting both regional and global climate change. Our best insight into how life in the high latitudes responded to this climatic shift is provided by the fossil record from Seymour Island, near the eastern coast of the Antarctic Peninsula. While extensive collections have been made from the La Meseta and Submeseta formations of this island, few avian taxa other than penguins have been described and mammalian postcranial remains have been scarce. Here, we report new fossils from Seymour Island collected by the Antarctic Peninsula Paleontology Project. These include a mammalian metapodial referred to Xenarthra and avian material including a partial tarsometatarsus referred to Gruiformes (cranes, rails, and allies). Penguin fossils (Sphenisciformes) continue to be most abundant in new collections from these deposits. We report several penguin remains including a large spear-like mandible preserving the symphysis, a nearly complete tarsometatarsus with similarities to the large penguin clade Palaeeudyptes but possibly representing a new species, and two small partial tarsometatarsi belonging to the genus Delphinornis. These findings expand our view of Eocene vertebrate faunas on Antarctica. Specifically, the new remains referred to Gruiformes and Xenarthra provide support for previously proposed, but contentious, earliest occurrence records of these clades on the continent.

Introduction

The Southern Hemisphere biota has been profoundly influenced by Mesozoic–Cenozoic continental breakup and climatic change. Before its fragmentation, the supercontinent Gondwana facilitated dispersal of terrestrial organisms between now-separated southern landmasses (Keast, 1972; Cracraft, 1973; Ali & Krause, 2011; Claramunt & Cracraft, 2015; Goin et al., 2016). Early discoveries suggest that Antarctica was central to this pattern of terrestrial movement, acting as a bridge between what is now South America and Australia (Woodburne & Zinsmeister, 1984; Zinsmeister, 1986). This widespread dispersal ended with the final breakup of Gondwana (Reguero et al., 2014, and reviewed by Torsvik & Cocks, 2013). Through the last part of this breakup, the Antarctic climate shifted from being warm and seasonally wet to increased periods of ice cover by the early to middle Eocene (Poole, Hunt & Cantrill, 2001; Ivany et al., 2011; Jacques et al., 2014); by the earliest Oligocene (∼33.9 Ma) Antarctica experienced complete glaciation (Zachos et al., 2001; Birkenmajer et al., 2004; Ivany et al., 2006; Barker, Diekmann & Escutia, 2007).

Insights into how the Antarctic biota was shaped by tectonic and climatic shifts have come from the upper Eocene La Meseta and Submeseta formations on Seymour Island (Marambio Island), the best-studied fossil vertebrate fauna from Antarctica (e.g.,  Reguero, Marenssi & Santillana, 2002; Reguero et al., 2014). This assemblage has been proposed to most closely resemble contemporaneous faunas from Patagonia (Reguero, Marenssi & Santillana, 2002) which were separated from what is now the Antarctic Peninsula by the flooding of the Weddellian Isthmus at the end of the Paleocene (Eagles & Jokat, 2014; Reguero et al., 2014). The fossil record of the La Meseta and Submeseta formations is famously dominated by stem penguins, including some of the tallest penguins that ever lived (Tambussi et al., 2006; Jadwiszczak, 2006; Tambussi & Acosta Hospitaleche, 2007; Acosta Hospitaleche, 2014; Acosta Hospitaleche & Reguero, 2014; Jadwiszczak & Mörs, 2019). The non-penguin vertebrate fossil record mostly comprises isolated teeth and bones representing an array of marsupial, gondwanathere, ungulate, cetacean, and other eutherian mammals as well as a possible non-therian dryolestoid (see Woodburne & Zinsmeister, 1982; Marenssi et al., 1994; Reguero, Marenssi & Santillana, 2002; Reguero & Gasparini, 2006; Case, 2006; Martinelli et al., 2014; Buono et al., 2016; Gelfo, Lopez & Santillana, 2017; Gelfo et al., 2019) as well as non-penguin birds (e.g., Tambussi & Acosta Hospitaleche, 2007; Jadwiszczak, Gazdicki & Tatur, 2008; Cenizo, 2012; Tambussi & Degrange, 2013; Cenizo, Acosta Hospitaleche & Reguero, 2015; Acosta Hospitaleche & Gelfo, 2015; Acosta Hospitaleche & Gelfo, 2017; and reviewed in Acosta Hospitaleche et al., 2019a). Here we report additional mammalian and avian specimens recovered from Antarctica by the Antarctic Peninsula Paleontology Project (AP3), including a gruoid and a xenarthran fossil, and discuss their biogeographic implications.

Geologic setting

The fossils here described were collected from Eocene deposits on Seymour Island. The island is located approximately 100 km east of the Antarctic Peninsula in the James Ross Basin and contains fossiliferous marine sedimentary units ranging from Late Cretaceous to late Eocene/earliest Oligocene in age (Sadler, 1988; Marenssi, Santillana & Rinaldi, 1998; Montes et al., 2013). These deposits can be divided into the Marambio Group (Santonian–Danian; Marenssi, Santillana & Rinaldi, 1998) and the unconformably overlying Seymour Island Group (Paleogene; Sadler, 1988; Marenssi, Santillana & Rinaldi, 1998). The Marambio Group is made up of the Santa Marta (Santonian to Campanian), Snow Hill Island (Campanian to Maastrichtian), López de Bertodano (Maastrichtian to Danian), and Sobral (Danian) formations, while the Seymour Island Group includes the Cross Valley (upper Paleocene), La Meseta (Eocene), and Submeseta (upper Eocene to lowermost Oligocene) formations (Sadler, 1988; Bowman et al., 2016; Montes et al., 2013).

The La Meseta Formation is predominantly made up of mudstones and sandstones and is interbedded with conglomerates (Marenssi, Santillana & Rinaldi, 1998). The formation was divided into numbered informal units called TELMs (Tertiary Eocene La Meseta) 1–7 by Sadler (1988). These units were later divided into allomembers (from bottom to top: Valle de Las Focas, Acantilados I, Acantilados II, Campamento, Cucullaea I, Cucullaea II, and Submeseta), with the Submeseta Allomember (comprising the upper portion of TELM 6 and all of TELM 7) later reassigned as the Submeseta Formation (Marenssi, Santillana & Rinaldi, 1998; Montes et al., 2013). The Submeseta Formation represents the late Eocene (Priabonian) to earliest Oligocene, between 43.4 and 33 Ma (Marenssi, Santillana & Rinaldi, 1998; Marenssi, 2006; Montes et al., 2013). This unit is composed predominantly of fine sandstones and mudstones from a shallow marine environment, but may reflect a sea level rise towards the top of the section (Marenssi, Net & Santillana, 2002; Marenssi, 2006; Montes et al., 2013). The Submeseta Formation itself has been divided into three units defined by their boundary discontinuities: the lower Submeseta I, middle Submeseta II, and upper Submeseta III allomembers (Marenssi, Santillana & Rinaldi, 1998; Montes et al., 2013).

Fossils were surface-collected at four localities on the northeastern end of Seymour Island (Fig. 1) from cross-bedded fine sandstone units interpreted as shallow marine or estuarine environments. A mammalian metacarpal was collected from locality S124, located in the Cucullaea I Allomember of the La Meseta Formation (52.8–49 Ma). A penguin mandible was collected from locality S074, within the Cucullaea II Allomember of the La Meseta Formation (49–45.9 Ma), and all tarsometatarsi were collected from the Submeseta I Allomember of the Submeseta Formation (43.4–41 Ma) at localities S123 and S117/122.

Figure 1 Map and geology of Seymour Island, James Ross Basin, Antarctic Peninsula, showing the localities where new material was recovered.

Fossil localities are marked by dots in the La Meseta and Submeseta formations. The mammalian metacarpal was recovered from S124, the penguin mandible from S074, and the three tarsometatarsi from S123 and S117/122. Modified from Montes et al. (2013).

Systematic Paleontology

MAMMALIA Linnaeus, 1758	
EUTHERIA Gill, 1872	
XENARTHRA Cope, 1889	
Gen. et sp. indet.	
(Fig. 2, Figure S1)	

Material—TMM 44190-1, left metacarpal II.

Locality—S124, Seymour Island, Antarctic Peninsula.

Formation/Age—Cucullaea I Allomember (TELM 4), La Meseta Formation, late Eocene.

Description—TMM 44190-1 is weathered and missing the distal epiphysis. The distal surface (Fig. 2F) is extensively pitted and shows no sign of breakage, indicating that TMM 44190-1 likely belonged to a juvenile individual. It has a maximum proximodistal length of 33 mm as preserved and a maximum mediolateral width of 21 mm at both the proximal and distal ends.

Figure 2 Left metacarpal II (TMM 44190-1) referred to Xenarthra.

(A) lateral; (B) medial; (C) palmar; (D) dorsal; (E) proximal; and (F) distal views. Features are tentatively indicated, as they are difficult to assign given the isolated and fragmentary nature of the element. Abbreviations: ?mcc, facet for either the metacarpal-carpal complex or for metacarpal I; ?mcIII, probable facet for metacarpal III; ?t, trapezoid facet. Scale bar = 10 mm.

In medial view, the articular facet for what could be the metacarpal-carpal complex or for metacarpal I is sharply- defined, forming the proximal part of the palmar margin and projecting well- palmarly of the rest of this margin (Fig. 2A). The remainder of the medial face is marked by two rugosities: one at the proximopalmar part and the other across the entire distal half (Fig. 2A). These rugosities are separated by a smooth sulcus, resulting in a notched medial margin in dorsal/palmar views (Figs. 2C–2D). In lateral view, an articular facet, potentially for metacarpal III, is rugose and worn (Fig. 2B). Due to the bone being hourglass-shaped in medial and lateral views (Figs. 2A–2B) but sub rectangular in dorsal and palmar views (Figs. 2C–2D), the latter faces are broadly concave and saddle-shaped. The articular face for the ?trapezoid carpal is triangular with sharply-defined medial and dorsal margins, and slopes slightly towards the medial edge (Fig. 2E).

Comparisons—The majority of mammal fossils from the Eocene of Seymour Island comprise teeth and isolated postcranial material of marsupials (Woodburne and Zinsmeister, 1982; Goin & Carlini, 1995; Goin et al., 1999; Goin et al., 2007; Chornogubsky, Goin & Reguero, 2009; Goin et al., 2018), Astrapotheria (Bond et al., 2011), Gondwanatheria (Goin et al., 2006; Gelfo et al., 2015), South American native ungulates (Litopterna, Bond, Reguero & Vizcaíno, 2006; Gelfo et al., 2015), and potential xenarthrans (Marenssi et al., 1994; Vizcaíno & Scillato-Yané, 1995). These mammalian records were most recently reviewed by Gelfo et al. (2019). Given the juvenile status and isolated nature of TMM 44190-1, as well as the relative paucity of described mammalian metacarpals from this time period, it is difficult to make comparisons with other contemporaneous fossils. We therefore limit detailed comparison to clades known from Seymour Island, except for gondwanatheres for which no metacarpal material has been described.

The metacarpals of Paleogene marsupials are more elongate and gracile than TMM 44190-1, with more rounded proximal ends. Metacarpal II specifically shows less prominent articular facets for metacarpal III compared to the new fossil (see Forasiepi et al., 2014: fig. 12a–c). In astrapotheres, the second metacarpal is subrectangular and apparently dorso-palmarly compressed while widening distally, but has a prominent ridge along the lateral side that is not seen in the new fossil (Scott, 1937: plate V, fig. 1). In ?Parastrapotherium the proximal facet appears somewhat flattened similar to the condition in TMM 44190-1, but the metacarpal contains a shallow pit along the medial surface not seen in the new fossil. The metacarpal also does not taper as dramatically towards the midpoint of the element (Scott, 1909: fig. 6). However, there are no detailed illustrations of individual metacarpals of ?Parastrapotherium, and so these apparent similarities are difficult to further assess. South American native ungulates metacarpals are more elongated than the new fossil, have a saddle-shaped articulation for the trapezoid, and have more points of articulation with the carpals than are seen on TMM 44190-1 (Shockey & Flynn, 2007: fig. 4).

Overall the robustness and proportions of TMM 44190-1 are most consistent with the metacarpals of xenarthrans, though the specimen does not have conspicuous affinities with a particular subclade. Within Cingulata (glyptodonts, armadillos), the metacarpals and metatarsals are most often figured in articulated posture which complicates comparisons, especially of their articular facets (see Gillette & Ray, 1981; Fernicola et al., 2018; Cuadrelli et al., 2019; Scott, 1903–1905). The second metacarpals of the North American Glyptotherium texanum and South American Glyptodon reticulatus are of similar proportion and share the hourglass configuration in lateral and medial views as TMM 44190-1, though they have less dorso palmar tapering towards the center of the diaphysis than is seen in the new fossil (Gillette & Ray, 1981: fig. 33; Cuadrelli et al., 2019: fig. 5c). The articulation for the trapezoid in both taxa is dramatically concave, in contrast to the flat, triangular surface in TMM 44190-1, and there are two points of contact along the proximal end between metacarpals II and III rather than one. The articular surface for metacarpal III in the Miocene armadillo relative Proeutatus is similarly shaped to that of TMM 44190-1, though it is only figured as articulated and this cannot be confirmed (Scott, 1903–1905: plate XIV, figs. 4 and 5). The Oligocene genus Peltephilus is most similar amongst representatives of the cingulates to the new fossil, though it appears longer (Scott, 1903–1905: plate XVI, fig. 11). The dorsal edge of the articular surface for the trapezoid appears similarly flat and sloped as in TMM 44190-1, and the facets for metacarpal III and the metacarpal-carpal complex appear proportional to the new fossil as well (Scott, 1903–1905: plate XVI, fig. 11). However towards the distal end the metacarpal tapers in a way that TMM 44190-1 would not, even if the missing epiphysis were present.

The new fossil also shares some features seen in Pilosa (anteaters, sloths). Although no second metacarpals have been recovered for Paleogene folivorans (sloths), there are similarities with the metacarpals of younger sloth taxa. The overall shape of the metacarpal resembles that of adult specimens of Megalonyx spp. (TMM 30967-1845; see also De Iuliis & Cartelle, 1999: fig. 7b), Hapalops spp. (Stock, 1925: fig. 23), and Eucholeops spp. (De Iuliis et al., 2014: figs. 9 and 10). TMM 44190-1 is more robust than the metacarpal II of some other xenarthrans such as Thalassocnus (Amson et al., 2015) and Mionothropus cartellei (De Iuliis, Gaudin & Vicars, 2011: fig. 11), but more closely matches the proportions of Pleistocene ground-dwelling taxa such as Megatherium urbinai (Pujos & Salas, 2004). At the distal end of the metacarpal, TMM 44190-1 is more mediolaterally wide than it is dorsopalmarly tall, a proportion that does not match other measured sloth metacarpals (e.g., see Eucholeops; De Iuliis et al., 2014; Amson, De Muizon & Gaudin, 2017: character 20). The ratio of proximodistal length to dorsopalmar depth of TMM 44190-1 is 1.57, which falls far below most values reported for other sloth taxa and near Glossotherium, well within the range considered stout (Amson, De Muizon & Gaudin, 2017: appendix S3). The trapezoid facet is sub-planar, similar to that of Pseudolestodon hexaspondylus and Simomylodon uccasamamensis (Haro, Tauber & Krapovickas, 2017: character 334). However, this is different than the concave facets seen in other sloths such as Eremotherium eomigrans (De Iuliis & Cartelle, 1999: fig. 10c) and Thalassocnus (Amson et al., 2015: figs. 34 and 35). In TMM 44190-1, the articular facet for the metacarpal-carpal complex does not extend distally to the midpoint of the shaft as in younger forms such as Hapalops (Miocene) and Nothrotherium (Pleistocene, Stock, 1925). The new fossil is also lacking facets seen in some sloth taxa, such as one for the magnum as in E. eomigrans, Megatherium americanum, and Thalassocnus (De Iuliis & Cartelle, 1999: fig. 7; Amson et al., 2015: fig. 35; see Amson, De Muizon & Gaudin, 2017: character 16) or the unciform as seen in Scelidotherium (Cuenca Anaya, 1995: p. 176). It is unclear if the rugose surface texture along the medial face is consistent with its juvenile status or could be due to a closely-appressed digit I (as in for example Mionothropus; De Iuliis, Gaudin & Vicars, 2011). There are no figured second metacarpals for extinct Vermilingua (anteaters), but in extant anteaters the second metacarpals are long and thin and quite different from the new fossil (Beddard, 1902: fig. 95; Orr, 2005: fig. 4b).

Being Eocene in age, and therefore close to the estimated early radiation of xenarthran groups, the likelihood of a single metacarpal having a diagnostic characteristic of one particular xenarthran subclade may be expected to be low. The fossil shows several features seen in extinct members of both Pilosa and Cingulata, and particularly with early diverging examples such as Peltephilus. While these similarities suggest TMM 44190-1 is a xenarthran, further subclade attribution is not possible.

AVES Linnaeus, 1758	
NEOGNATHAE Pycraft, 1900	
GRUIFORMES Bonaparte, 1854 sensuHackett et al., 2008	
?GRUOIDEA Vigors, 1825 sensuClarke, Norell & Dashzeveg, 2005	
Gen. et sp. indet.	
(Fig. 3)	

Material—TMM 44189-2, distal end of left tarsometatarsus.

Locality—S123, Seymour Island, Antarctic Peninsula.

Formation/Age—Submeseta I Allomember (TELM 7), Submeseta Formation, late Eocene.

Description—TMM 44189-2 preserves the bases of trochleae II–IV as well as the dorsal and plantar openings of the distal vascular foramen. The maximum mediolateral width as preserved is 25 mm (Fig. 3). The distal vascular foramen is proximodistally elongate in dorsal and plantar views (Figs. 3A–3B), and the dorsal opening of the foramen is set in a deep sulcus (Fig. 3A). The plantar opening of the foramen is slightly lateral to and near the midline and is positioned distal to the juncture of trochleae II and III (Fig. 3B). The fossa for the m. supratrochlearis plantaris is shallow (Fig. 3B). Most of trochlea II is missing but appears to lack a well-defined plantar crest extending proximally from the ala of the trochlea (Fig. 3B). In distal view, trochlea II is plantarly deflected, and trochleae III and IV are widely spaced (Fig. 3E).

Figure 3 Distal end of a left tarsometatarsus (TMM 44189-2) compared to those of extant gruiforms.

The fossil (A–E) and comparative materials (F–Y) shown in dorsal, plantar, medial, lateral, and distal views. Abbreviations: II, trochlea II; III, trochlea III; IV, trochlea IV; dvf, distal vascular foramen. Scale bar = 10 mm.

Comparison—A portion of a tarsometatarsus (MLP 90-I-20-9) recovered from Seymour Island was previously figured as gruiform but not described (Tambussi & Degrange, 2013: fig. 6.1g). This specimen comprises a distal diaphysis that is broken proximal to the trochleae, of which only the proximal most part of trochlea IV is preserved. Despite the partial preservation, this fossil does not clearly show the splayed trochlear arrangement present in extant Gruoidea (cranes, trumpeters, and limpkins) and in TMM 44189-2. The lack of published measurements or description of MLP 90-I-20-9 make comparisons with the new fossil difficult, but as figured it appears that this fossil is narrower than TMM 44189-2. Further evaluation is needed to determine the exact relationship between the two fossils, but TMM 44189-2 exhibits a suite of character states that allows for a more detailed assessment.

Phylogenetic studies that have recovered a monophyletic Gruiformes (sensu Hackett et al., 2008) have not recovered synapomorphies from the distal tarsometatarsus that can be assessed in this specimen (see Mayr & Clarke, 2003; Musser & Cracraft, 2019; Musser, Ksepka & Field, 2019). However, TMM 44189-2 presents a combination of character states most consistent with Gruiformes: (cranes, rails, and allies): (1) trochleae III and IV projecting well distal of II; (2) plantar deflection of trochlea II (as inferred from the base and preserved ala); (3) trochlea III positioned dorsal to trochlea IV in distal view; (4) dorsoplantar flattening and mediolateral broadening of the supratrochlear region; (5) position of the distal vascular foramen near the midline and away from the lateral margin in plantar view; and (6) wide spacing of trochleae, III and IV. Within Gruiformes, TMM 44189-2 is more similar to Gruoidea than Ralloidea (rails, finfoots, and flufftails) based on the following characteristics: (1) trochlea II is not as plantarly deflected in the new fossil as in ralloids; (2) trochlea II projects farther distally relative to III and IV in the new fossil than in ralloids; (3) the distal vascular foramen is located midway between the midline and the lateral margin in plantar view, unlike in ralloids where it is located on the midline; (4) the supratrochlear region of the new fossil is mediolaterally broader and trochleae II and IV are more widely spaced than in ralloids; (5) the distal margin of the distal vascular foramen is in line with the proximal extent of trochlea III in dorsal view, unlike in ralloids; (6) the distal vascular foramen is located closer to the intertrochlear incisure in plantar view than in ralloids; and (7) in dorsal view, the proximal extents of trochleae III and IV are subequal, whereas IV is proximal to III in ralloids.

The morphology of TMM 44189-2 is not unambiguously consistent with any particular gruoid subclade. Trochlea II is not as plantarly deflected as in Gruidae (cranes) or Aramus guarauna (limpkin), and is more like the condition observed in Psophiidae (trumpeters). As in A. guarauna and Gruidae, trochlea III is the most dorsally positioned trochlea. There is a shallow depression at the plantar base of trochlea IV along the beginning of an ala that is most like that of Balearica pavonina and Grus canadensis among compared Gruiformes. However, the trochlear bases of TMM 44189-2 are not as dorsoventrally thick as those of A. guarauna and Gruidae and are more like those of Psophia viridis. The distal vascular foramen of TMM 44189-2 is ovoid in plantar view, with the long axis at an oblique angle to the long axis of the shaft, as in Gruidae but unlike A. guarauna and Psophia. In dorsal view, the distal vascular foramen is set in a broad, shallow sulcus as in B. pavonina and Psophiidae; by contrast, this sulcus is deep and sharply defined in Gruoidea. The fossil lacks the sharp plantar crest extending proximally from the ala of trochlea II observed in Gruoidea. A marked, circular depression is located between trochleae II and III, and is most like the condition in Psophia, A. guarauna, and B. pavonina, although the observed depth may be an artifact of preservation.

SPHENISCIFORMES Sharpe, 1891 sensuClarke, Olivero & Puerta, 2003	
Delphinornis sp.	
(Figs. 4A–4H)	

Material—TMM 44189-1, left tarsometatarsus.

Locality—S123, Seymour Island, Antarctic Peninsula.

Formation/Age—Submeseta I Allomember (TELM 7), Submeseta Formation, late Eocene.

Description—TMM 44189-1 (Figs. 4A–4D) is missing its proximal end and trochlea IV. It is the more complete example of the two tarsometatarsi (including TMM 44188-2, described below) recovered by the 2016 AP3 expedition that represent a small-bodied penguin. The specimen is similar in size to the tarsometatarsus of the extant Spheniscus humboldti (Humboldt penguin). The hypotarsal crests are not preserved; however, an abraded surface appears to mark the former distal most extent of the medial hypotarsal crest. The medial proximal vascular foramen is positioned directly medial to the abraded surface that potentially corresponds to the medial hypotarsal crest. The distal vascular foramen is partially preserved, including a distinct plantar foramen just lateral to the trochlea of metatarsal II.

Comparison—TMM 44189-1 is referable to Sphenisciformes (penguins) based on its overall proportions, morphology, and extreme osteosclerosis. The specimen possesses both intertarsal grooves (Fig. 4A), unlike the much larger penguin tarsometatarsus described below (TMM 44188-1). Although apparent, the medial intertarsal groove is shallower than that of all comparable extant species. The lateral intertarsal groove is present and deep, similar to the condition in extant species as well as the extinct taxa Delphinornis, Marambiornis, and Mesetaornis (Myrcha et al., 2002; Jadwiszczak & Mörs, 2019). The groove does not taper strongly distally, as in Marambiornis and Mesetaornis. The trochlea of metatarsal II is positioned more medially than those of all comparable extant penguin species (Figs. 4A–4B and 4D), resulting in a wide medial intertrochlear incisure that appears most similar to that of Delphinornis (Myrcha et al., 2002; Jadwiszczak & Mörs, 2019). A distal vascular foramen is present (Figs. 4A–4B) as in taxa from the Paleocene of New Zealand, including Muriwaimanu, as well as the small Antarctic taxa Delphinornis, Marambiornis, and Mesetaornis (Myrcha et al., 2002; Chávez Hoffmeister, 2014; Jadwiszczak, 2015; Jadwiszczak & Mörs, 2019). The distally-opening passage of the m. extensor brevis digiti IV is confluent with the distal vascular foramen (Fig. 4A), as in Delphinornis, Marambiornis, and Mesetaornis (Myrcha et al., 2002; Hoffmeister, 2014; Jadwiszczak, 2015; Jadwiszczak & Mörs, 2019). The plantar opening of the distal vascular foramen is more distally positioned (Fig. 4B) than in Marambiornis and Mesetaornis and is similar in morphology to that of Delphinornis (Myrcha et al., 2002; Jadwiszczak & Mörs, 2019). Based on these traits, we assign this fossil to Delphinornis. Tarsometatarsal traits that distinguish among the three species of Delphinornis, namely morphology of the intercotylar eminence, medial hypotarsal crest, proximal vascular foramina, and relative sizes of all three trochleae (Myrcha et al., 2002), are not preserved in this specimen.

Figure 4 New sphenisciform fossil material.

A small left tarsometatarsus (TMM 44189-1) in (A) dorsal, (B) plantar, (C) proximal, and (D) distal views; a small left tarsometatarsus (TMM 44188-2) in (E) dorsal, (F) plantar, (G) proximal, and (H) distal views; a large left tarsometatarsus (TMM 44188-1) in (I) dorsal, (J) plantar, (K) proximal, and (L) distal views; and a partial mandible (TMM 44187-1) in (M) dorsal, (N) ventral, (O) right (with associated material), and (P) left (with associated material) views. Abbreviations: II, trochlea II; III, trochlea III; dvf, distal vascular foramen; es, extensor sulcus; fs, flexor sulcus; h, hypotarsus; ia, intercotylar area; ie, intercotylar eminence; lc, lateral cotyla; mc, medial cotyla; ms, mandibular symphysis; pvf, proximal vascular foramen. Scale bar = 10 mm.

Material—TMM 44188-2, left tarsometatarsus.

Locality—S122, Seymour Island, Antarctic Peninsula.

Formation/Age—Submeseta I Allomember (TELM 7), Submeseta Formation, late Eocene.

Description—TMM 44188-2 (Figs. 4E–4H) is missing the proximal end and all three trochleae. It is the less complete of the two 2016 specimens that represent a small penguin morphotype. It is comparable in size to TMM 44189-1 and identical to that specimen in all preserved morphologies. Therefore, it likely also represents a fragmentary specimen of the clade Delphinornis.

Palaeeudyptes sp.	
(Figs. 4I–4L)	

Material—TMM 44188-1, left tarsometatarsus.

Locality—S117, Seymour Island, Antarctic Peninsula.

Formation/Age—Submeseta I Allomember (TELM 7), Submeseta Formation, late Eocene.

Description—TMM 44188-1 is a mostly complete tarsometatarsus that is missing trochlea IV (Figs. 4I–4L). It has a proximodistal length of 45 mm and proximal mediolateral width of 39 mm. The medial and lateral proximal cotyla are separated dorsally by a pronounced intercotylar eminence and plantarly by a planar intercotylar area. The medial proximal vascular foramen is positioned just distal to the distal terminus of the medial hypotarsal crest, and is more developed than the lateral proximal vascular foramen. A scar for the m. tibialis cranialis is present on the dorsal face as a short ridge that extends distally from the proximal margin. The tarsometatarsus lacks an appreciable medial dorsal intertarsal sulcus but exhibits a lateral sulcus.

Comparison—TMM 44188-1 most closely resembles the tarsometatarsus of the contemporaneous Seymour Island penguin Palaeeudyptes gunnari based on the following features: (1) a concave medial margin, (2) a medial proximal vascular foramen that is larger than the lateral vascular foramen, (3) absence of an osseous ridge from the intermediate hypotarsal crest to the medial margin as in P. antarcticus (Myrcha et al., 2002), and (4) a proximally-positioned scar for the m. tibialis cranialis. The new specimen can be differentiated from the contemporaneous and similarly sized Archaeospheniscus, known from Seymour Island and New Zealand, based on the lack of a medial dorsal intertarsal sulcus and unequally sized proximal vascular foramina (Simpson, 1971b; Myrcha et al., 2002). The new specimen is also distinct from the contemporaneous Anthropornis, known from Seymour Island and New Zealand, in which the scar for the m. tibialis cranialis is positioned more distally and the medial proximal vascular foramen is larger than the lateral (Myrcha et al., 2002). Lastly, TMM 44188-1 is significantly smaller than P. gunnari, P. klekowskii, and the two species of Palaeeudyptes known from New Zealand, P. antarcticus and P. marplesi (Simpson, 1971a; Myrcha et al., 2002), and may therefore represent a new species within Palaeeudyptes.

Gen. et sp. indet.	
(Figs. 4M–4P)	

Material—TMM 44187-1, rostral portion of mandible with associated fragments of caudal rami.

Locality—S074, Seymour Island, Antarctic Peninsula.

Formation/Age—Cucullaea II Allomember (TELM 5), La Meseta Formation, late Eocene.

Description—TMM 44187-1 comprises the rostral end of a mandible that includes most of the symphyseal region, with the left mandibular ramus being more complete than the right (Figs. 4M–4P). The rostralmost tip is missing. The preserved portion of the left ramus measures 171 mm in length and 7 mm in maximum width. An additional fragment of this ramus measures 81 mm in length, demonstrating that, when complete, the left mandibular ramus was at least 252 mm in length. However, the articular regions of the mandible are missing, indicating that the original length of the bone was even greater. The preserved portion of the symphysis measures 37 mm in length and 10 mm wide at its rostrocaudal midpoint. We estimate the length of the complete symphysis at 40 mm. The mandible is slender and pointed but sturdily constructed, and is excavated by vascular canals throughout much of its length. The tip of the mandible is straight, and the rami meet the symphysis along a straight line rather than at an angle. Mandibular fossae are not preserved.

Comparison—Few penguin mandibles have been reported from Seymour Island, and to date only one has been referred to a known species (MLP 14-XI-27-84, assigned to Anthropornis grandis; Acosta Hospitaleche et al., 2019b). TMM 44187-1 has a shorter symphysial region than that reported for MLP 14-XI-27-84 (∼45 mm) and lacks the ventrally convex, dorsally concave condition seen in A. grandis (Acosta Hospitaleche et al., 2019b). TMM 44187-1 differs from other partial mandibles described from Seymour Island (MLP 91-II-4-221, MLP 92-II-2-195, IB/P/B-0653; Jadwiszczak, 2006; Acosta Hospitaleche & Haidr, 2011), which are more tapered towards the distal end and have thinner rami. However, the overall morphology of the symphysis is comparable to other Seymour Island fossils described despite differing in overall dimensions (MLP 96-I-6-48, MLP 78-X-26-144, IB/P/B0617e, MLP 14-XI-27-27; Jadwiszczak, 2006; Acosta Hospitaleche & Haidr, 2011; Jadwiszczak, 2011; Haidr & Acosta Hospitaleche, 2017). The fossil MLP 96-I-6-48 has vascular pitting and a flattened dorsal surface similar to those seen on TMM 44187-1 (Acosta Hospitaleche & Haidr, 2011), and the pitting is consistent with the morphology of extant adult Aptenodytes forsteri (Emperor penguin; Sosa & Acosta Hospitaleche, 2018). The shape of the rami of MLP 96-I-6-48, MLP 78-X-26-144, IB/P/B-0617e, and MLP 14-XI-27-27 are all similar to that of TMM 44187-1 (Acosta Hospitaleche & Haidr, 2011; Jadwiszczak, 2011; Haidr & Acosta Hospitaleche, 2017). An unassigned, mostly complete mandible with associated maxilla most closely resembles the new fossil; MLP 14-XI-27-27 appears to have similarly straight mandibular rami and appears to have a flattened dorsal surface (Haidr & Acosta Hospitaleche, 2017: fig. 4b). However, only a dorsal view is figured and the specimen lacks formal description, making it difficult to determine if the two fossils are from the same taxon. These fossils pertain to Paleogene penguins with spear- or dagger-like bills characteristic of stem species (Slack et al., 2006; Clarke et al., 2007; Clarke et al., 2010; Ksepka & Clarke, 2010; Jadwiszczak, 2011; Haidr & Acosta Hospitaleche, 2012; Acosta Hospitaleche et al., 2019b). However due to the new fossil’s differences to assigned mandibular material (Acosta Hospitaleche et al., 2019b) and its partial preservation, TMM 44187-1 is not considered referable to any known Eocene taxon at this time.

Discussion

Though our recently collected Eocene material is fragmentary, it provides additional support for records of the presence of mammalian and avian taxa previously proposed from even more fragmentary and controversial single elements. These new records are also consistent with those expected for the Eocene of Antarctica given longstanding hypotheses of a biotic connection between Antarctica and South America during the Paleogene as well as penecontemporaneous fossil discoveries from Patagonia (see Reguero, Marenssi & Santillana, 2002; Sallaberry et al., 2010; Yury-Yáñez et al., 2012; Acosta Hospitaleche & Olivero, 2016; Reguero et al., 2014). The Eocene mammalian record otherwise comprises gondwanatheres, marsupials, cetaceans, ‘South American native ungulates’ (e.g., a litoptern, astrapotheres), and additional, enigmatic eutherians (Woodburne & Zinsmeister, 1984; Borsuk-Bialynicka, 1988; Case, Woodburne & Chaney, 1988; Bond et al., 1990; Hooker, 1992; Marenssi et al., 1994; Bargo & Reguero, 1998; Fostowicz-Frelik, 2003; Reguero & Gasparini, 2006; Case, 2006; Reguero et al., 2013; Gelfo et al., 2015; Buono et al., 2016; reviewed in Gelfo et al., 2019). Indeed, in addition to the described metacarpal, the 2016 AP3 expedition recovered a vertebra (from locality S123) consistent with referral to a basilosaurid archaeocete. Although new collections improve our understanding of biodiversity on Antarctica during the Eocene, they also highlight the need to recover and describe more material to elucidate a nuanced understanding of biotic exchange during this key time period.

Previous reports of xenarthrans from the Eocene of Seymour Island—based on a distal ungual phalanx and an incomplete tooth—were initially assigned to Tardigrada (= Folivora) (Marenssi et al., 1994; Vizcaíno & Scillato-Yané, 1995), but were later questioned (Bargo & Reguero, 1998; MacPhee & Reguero, 2010). The phalanx was recovered from the Cucullaea I Allomember, approximately 0.7 km away from where TMM 44290-I was collected, but lacks formal description and reportedly has been lost, precluding reevaluation (Bargo & Reguero, 1998; figured in Gelfo et al., 2019: fig. 5b). The phalanx was noted to be indistinguishable from the earliest known Vermilingua (anteater) fossil from Patagonia, and based on histological study the tooth was reassigned to Mammalia indet. (MacPhee & Reguero, 2010). Therefore, the newly described metacarpal is either further evidence, or new evidence, that Xenarthra was indeed present on Antarctica during the Eocene depending upon one’s stance with regard to prior controversies. This record is consistent with the estimated timing of origin for Folivora by the early Eocene, and of Xenarthra in the Paleocene (e.g., Presslee et al., 2019).

Xenarthra is proposed to have originated in South America, and thus is plausibly anticipated in the Paleogene of Antarctica given inferred land connections between these continents during the early Cenozoic (Woodburne & Case, 1996; Delsuc et al., 2019; Presslee et al., 2019). The new metacarpal extends the known Paleogene geographic range of Xenarthra into Antarctica. Xenarthran limb bones and osteoderms have been reported from the early Eocene (55–50 Ma) of Brazil (Gaudin & Croft, 2015; Superina & Loughry, 2015), but the earliest reported members of Pilosa date to 31.5 Ma in Chile and Argentina (McKenna, Wyss & Flynn, 2006; Gaudin & Croft, 2015) while described early cingulates from Patagonia are potentially early Eocene (Marshall, Hoffstetter & Pascual, 1983; Croft, Flynn & Wyss, 2007). If pilosan, the new material would indicate that this clade was present in Antarctica by at least 35 Ma, four million years before it is known in South America. A cingulate affinity would support the presence of the group in both Antarctica and southern South America around the same time. However, the paucity of other described Paleogene xenarthran postcranial material limits definitive analysis of the phylogenetic affinities and ecology of this individual.

Antarctic bird fossils from non-penguins are rare, and only a few have been named as species. They account for less than half of known extinct avian species diversity on the continent (Tambussi & Acosta Hospitaleche, 2007; Tambussi & Degrange, 2013; reviewed in Acosta Hospitaleche et al., 2019a), but comprise an even smaller fraction of unnamed material in collections. Therefore, the distal tarsometatarsus, although fragmentary, expands our understanding of Antarctic avian diversity during the late Eocene. A proposed gruiform from Seymour Island was previously figured (Tambussi & Degrange, 2013: fig. 6.1g) but its relation to the new fossil is difficult to assess. The new fossil exhibits preserved characters that allow for a more confident referral to core-Gruiformes, providing new evidence for the presence of the clade in Antarctica.

Our understanding of the paleobiogeography of Gruoidea remains incomplete due to a near lack of known remains of Gruidae from the Paleogene of the Southern Hemisphere and of reported parts of stem Aramidae and Psophiidae from the Paleogene (Mayr, 2009; Mayr et al., 2017; Musser & Cracraft, 2019). Of these three clades, Gruidae has the most extensive fossil record, with Eocene fossils primarily restricted to the Northern Hemisphere (e.g., Wetmore, 1933; Wetmore, 1940; Cracraft, 1969; Cracraft, 1973; Clarke, Norell & Dashzeveg, 2005; Mayr, 2009; Mayr, 2016). The new tarsometatarsus cannot confidently be referred to a subgroup within Gruoidea, and as such has different biogeographic implications depending on its affinities. If more closely related to Psophiidae or Aramidae, the new record suggests that these largely South American gruoid families were more broadly distributed at least as far back as the late Eocene and supports hypotheses of a distribution across Antarctic landmasses (Cracraft, 1982; Claramunt & Cracraft, 2015; Musser & Cracraft, 2019). If placed within Gruidae, the new tarsometatarsus could suggest that the gruid radiation may have been multi-directional; one radiation of cranes could have dispersed from North America to Eurasia via the Bering Land Bridge during the early Eocene and then dispersed towards west Eurasia over time (Claramunt & Cracraft, 2015), and another radiation could have arrived in Antarctica by the late Eocene via South America. However, more fossils are needed in order to gain a better understanding of the biogeography of this group and core-Gruiformes as a whole within the Southern Hemisphere.

The penguin mandible described here adds to the record of spear-billed penguins reported from the Eocene of Antarctica. Although fossil penguin cranial material is rare from Seymour Island, two beak morphotypes are known: long and narrow, spear-like shapes (proposed to indicate a primarily piscivorous diet) and shorter, broad morphs (proposed to indicate feeding on small crustaceans; Ksepka & Clarke, 2010; Acosta Hospitaleche & Jadwiszczak, 2011; Haidr & Acosta Hospitaleche, 2012; Haidr & Acosta Hospitaleche, 2017; Acosta Hospitaleche et al., 2019b). The shape of the mandible is consistent with a spear-billed morphology seen in other Antarctic remains (Acosta Hospitaleche & Jadwiszczak, 2011; Haidr & Acosta Hospitaleche, 2017; Acosta Hospitaleche et al., 2019b) and similar to those of penecontemporaneous species from Peru (Perudyptes devriesi, mid-Eocene; Clarke et al., 2007; Icadyptes salasi, mid-to late Eocene; Clarke et al., 2007; Ksepka et al., 2008) as well as Paleocene penguins from New Zealand (Muriwaimanu tuatahi; Sequiwaimanu rosieae: Slack et al., 2006; Ksepka & Clarke, 2010; Mayr et al., 2018). The morphology of the mandible is consistent with the spear-billed morphology typical of stem species (Clarke et al., 2007; Ksepka & Clarke, 2010; Haidr & Acosta Hospitaleche, 2012; Haidr & Acosta Hospitaleche, 2017; Acosta Hospitaleche et al., 2019b). Measurements of the symphysis and estimates of mandible length indicate that the individual represented by the new mandible would have been larger than the older New Zealand species Muriwaimanu tuatahi (Slack et al., 2006) and between size estimates reported for other Eocene Antarctic spear-bills recovered from Seymour Island (Acosta Hospitaleche & Jadwiszczak, 2011; Haidr & Acosta Hospitaleche, 2017; Acosta Hospitaleche et al., 2019b). The mandible does not reach the maximum mandibular length recorded for the South American Icadyptes salasi (Clarke et al., 2007), further supporting that a potential intermediate size class of these spear-billed taxa was present on Antarctica.

Penguins were diverse across the globe during the Eocene, with 14+ species described from Seymour Island alone (Jadwiszczak, 2006; Ksepka & Clarke, 2010; Acosta Hospitaleche, 2013 and references therein; Jadwiszczak & Mörs, 2017; Jadwiszczak & Mörs, 2019; reviewed in Acosta Hospitaleche et al., 2019a; Acosta Hospitaleche et al., 2019b). The materials described here add to our understanding of this diversity with new material from a range of size classes: one large spear-billed taxon, one medium-sized taxon represented by a tarsometatarsus, and one small taxon represented by two tarsometatarsi. It has been proposed that penguins were diverse in the mid- to late Eocene in part because of the increasing productivity of the southern oceans (Diester-Haass & Zahn, 1996; Clarke et al., 2007; Haidr & Acosta Hospitaleche, 2012; Villa, Fioroni & Persico, 2014). The range of body sizes and bill morphotypes observed have also been hypothesized to be the result of increased interspecific competition and size-based resource partitioning (Ksepka et al., 2008; Ksepka & Clarke, 2010; Haidr & Acosta Hospitaleche, 2012). The morphological diversity reported here may lend further support to these hypotheses.

Conclusions

New records from Antarctica expand our understanding of tetrapod biodiversity on the continent during the Eocene and support previously controversial reports of Gruiformes and Xenarthra. A metacarpal is proposed to possibly represent a new early record of Cingulata or Folivora, and lends support to previously reported xenarthran materials that have been subsequently questioned or lost. A new tarsometatarsus supports the presence of Gruiformes in Antarctica during the Eocene, adding to our understanding of the avian fossil record of Seymour Island. Newly reported penguin remains, including a spear-shaped mandible and three tarsometatarsi, add to the diversity of penguins known from this time. The nature of the Antarctic fossil record is characterized by isolated elements and is dominated by penguins, making new discoveries vital to furthering our understanding of diversity during a period of climate change and tectonic shifts. While historically fragmentary, new material from Antarctica is needed to elucidate trends in biodiversity and biotic exchange during a key episode of Earth history.

Supplemental Information

Figure S1 3D PDF of the left metacarpal II (TMM 44190-1) referred to Xenarthra

Click here for additional data file.

We gratefully acknowledge the assistance of all members of the Antarctic Peninsula Paleontology Project (AP3, http://antarcticdinos.org/) who participated in the 2016 field season. We are indebted to S. Shelley, J. Wible, E. Amson, R. MacPhee, A. Kramarz, and M. Lorente for comments on the mammalian material. We thank M. Brown and C. Sagebiel for curatorial assistance and specimen access, D. Wagner for fossil preparation, V. De Pietri for photos of MLP 90-I-20-9, and P. O’Connor, L. English, and H. McDonald for discussion and comments on this work. We thank A. Farke, E. Amson, and an anonymous reviewer for thorough and constructive reviews that improved this manuscript.

Institutional abbreviations

FMNH Field Museum of Natural History, Chicago, Illinois, USA

IB/P/B Andrzej Myrcha University Museum of Nature, Białystok, Poland

MLP Museo de La Plata, La Plata, Buenos Aires Province, Argentina

TMM Jackson School of Geosciences, Vertebrate Paleontology Laboratory, Austin, Texas, USA

Additional Information and Declarations

Competing Interests

Author Contributions

Field Study Permissions

Data Availability

Julia A. Clarke is an Academic Editor for PeerJ.

Sarah N. Davis conceived and designed the experiments, performed the experiments, analyzed the data, prepared figures and/or tables, authored or reviewed drafts of the paper, approved the final draft.

Christopher R. Torres conceived and designed the experiments, performed the experiments, analyzed the data, contributed reagents/materials/analysis tools, prepared figures and/or tables, authored or reviewed drafts of the paper, approved the final draft.

Grace M. Musser performed the experiments, analyzed the data, authored or reviewed drafts of the paper, approved the final draft.

James V. Proffitt performed the experiments, analyzed the data, prepared figures and/or tables, authored or reviewed drafts of the paper, approved the final draft.

Nicholas M.A. Crouch performed the experiments, analyzed the data, authored or reviewed drafts of the paper, approved the final draft.

Ernest L. Lundelius performed the experiments, analyzed the data, authored or reviewed drafts of the paper, approved the final draft.

Matthew C. Lamanna analyzed the data, contributed reagents/materials/analysis tools, authored or reviewed drafts of the paper, approved the final draft.

Julia A. Clarke conceived and designed the experiments, performed the experiments, analyzed the data, contributed reagents/materials/analysis tools, authored or reviewed drafts of the paper, approved the final draft.

The following information was supplied relating to field study approvals (i.e., approving body and any reference numbers):

Field work was approved by the National Science Foundation Office of Polar Programs.

The following information was supplied regarding data availability:

A 3D PDF of the mammalian metacarpal is available both as a Supplemental File and from MorphoSource: https://www.morphosource.org/Detail/ProjectDetail/Show/project_id/886 (TMM 44190-1).

All fossil material is accessioned at the University of Texas at Austin’s Texas Memorial Museum collection: TMM 44190-1, TMM 44189-2, TMM 44189-1, TMM 44188-2, TMM 44188-1, TMM 44187-1.

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
