# Peer review of "New mammalian and avian records from the late Eocene La Meseta and Submeseta formations of Seymour Island, Antarctica"

_PeerJ, doi:10.7717/peerj.8268_

## Round 0.1 · original submission · Major Revisions

The reviewers have provided a number of detailed and helpful comments, which suggest some significant revisions to the work, primarily in three areas:

1) The mammalian fossil bone and its identification should be reevaluated in light of the reviewer comments (see in particular notes from Amson, and the additional related comments from the anonymous reviewer on this topic).
2) Please consider the reviewer comments about stratigraphic nomenclature, particularly noting the relevant literature they mention. Aspects of the geological presentation should be updated accordingly.
3) Additional references and comparisons for the penguin material should be included, to ensure that this aspect of the work is up to date.

Also, one reviewer suggests it is not necessary to illustrate all of the penguin material given the work already present in the literature. I personally do not mandate removal of these photos (additional illustrations are nearly always useful, in my view), but leave it up to the authors.

·

Basic reporting

No comment

Experimental design

No comment

Validity of the findings

See my main comments.

Additional comments

Dear Editor, dear Authors,

I had the pleasure to review the manuscript (MS) titled New mammalian and avian records from the late Eocene La Meseta Formation of Seymour Island, Antarctica. It encloses the description of original material that potentially has extremely important consequences on our understanding of the evolution and biogeography of the clades in question. Indeed, vertebrate fossils from the Eocene of Antarctica are extremely rare. I therefore strongly believe that this manuscript will translate into an important paper with crucial data that have to be published. Nevertheless, I have to express my doubts regarding the attribution of the mammalian specimen to a sloth, as performed in the MS. I was not convinced by the arguments in favor of this attribution, neither was I convinced by the arguments used to discard an attribution to other potential clades. Personally, I do not think that I would be able to give a definitive attribution for the specimen under study, and would be more tentative about the conclusions that can be drawn from it. But again, I consider that the material is worth publishing, and hence strongly encourage the publication of the manuscript, after having considered the concern I detail below.
I have to make it clear that I am not qualified to review the parts of the manuscript that deal with the avian remains, so my comments only concern the mammal specimen. In addition to the comments given below, I am providing a pdf document showing tracked changes as well as an annotated figure at the end.

Introduction

Appropriate and well written.

Results
Major comments

1. The proportion of the metapodial (“short and broad”) is the main argument for its attribution to a sloth. I do not think that this is sufficient, as many other clades have metapodials with similar proportions. It would therefore be great to corroborate this with other arguments, such as the disposition of the articular facets and processes. The authors do discuss those, but I believe that strong differences remain with known folivorans. Moreover, I don’t agree with the identification of structures. A first problem comes from the fact that there is necessarily a mistake in the bone orientation, as figured in Fig. 2: if the bone is indeed a left McII, and the facet for the McIII correctly identified, then Fig. 2C should be a dorsal view (and 2D a palmar view), contrary to what the figure legend states. However, I would tend to agree with the fact that 2C is a palmar/plantar view (and 2D a dorsal view)…which means that the bone is not a Mc II and/or that the facet for the McIII is not correctly identified.
It is hard to be sure, but given the pictures and 3D model I have at hand, I would say that the view defined as the dorsal one (Fig. 2D) shows three contiguous facets. The whole facet that extends on the medial side of the bone is labelled as for the mcc. Again, hard to be sure, but given the angulation seen in dorsal view, I’d say that two bones articulate there, in addition to the proximal articular surface. I have annotated the Fig. 2 to illustrate this hypothesis.
The definition of the facet for the mcc is not entirely clear to me. Is it the entire proximal face of the bone that is assumed to articulate with the mcc? Based on the proximal view, I would assume (again it is hard to be sure of that) that two facets for distinct bones are visible (see my annotation on the figure). The text refers to a facet for the trapezoid, but it was not directly clear to me to what part of the bone it is referred to. It should be labelled on the Fig. 2E.
2. The trapezoid facet is described as “sub-planar, similar to that of Scelidotherium […].” But the disposition is completely different in Scelidotherium, in which the lateral part of the proximal articular surface articulates with the magnum. So the proximal surface as a whole is not sub-planar, as it is roughly the case in the specimen under study.
All in all, I am not convinced by the attribution to a sloth. The most resembling sloth metapodial I could find is the Mc III of Eucholeops (size seem to roughly match as well), but differences are still precluding me from taking a definitive stand (please note that I did not observe Eucholeops Mc II first hand, my assessment is based on the work of De Iuliis et al. 2014). The most important difference concerns the shape of the distal end of the diaphysis: in all the sloths I could think of, including Eucholeops, this end is dorsopalmarly deeper than mediolaterally wide, to accommodate well-developed dorsopalmar keel on the distal epiphysis. This argument could actually be used to rule out a folivoran affinity, because the preserved distal end of the metapodial under study is wider than deep. Such a morphology might indicate that the distal epiphysis was not keeled. Instead, it could have either accommodated a simple condyle, as in astrapotheres for instance, or a saddle-shaped articulation, like in glyptodonts. Note that I do consider that the overall morphology and size could fit that of the latter (see for instance figs 33 and 41 of Glyptotherium: Gillette and Ray 1981). One important difference remains, though, as the proximal (trapezoid?) facet is clearly concave in Glyptotherium, and flat in the new metapodial. I would not be able to completely rule out an armadillo metapodial either.

3. The section is concluded with the statement that the morphology of other mammals from the Formation, namely Astrapotheria, Gondwanatheria (note typo in the MS), and Litopterna, differ markedly from the specimen under study. I think it is important to expand this, as I am myself not entirely convinced. Keeping in mind that the distal end of the bone is missing, the bone is actually very reminiscent of the Mc II of ?Parastrapotherium as drawn in Scott (1909). Contrary to Glyptotherium, I would expect for the proximal facet of ?Parastrapotherium to be flat as well (I never studied first hand specimens of that taxon), as in the metapodial under study.
It is also unclear to me why a comparison with other xenarthrans was not performed.

More minor comments:

4. The authors seem to assume that there is necessarily a metacarpal-carpal complex (mcc) in sloths, but that’s not the case (e.g., Mionothropus, ;De Iuliis et al. 2011). Furthermore, the mcc can comprise different bones depending on the taxon. For instance, in Eremotherium, it includes the trapezoid (De Iuliis and Cartelle 1994), which seems to be assumed to be separate here. I would recommend to clarify the issue.
5. The comparison ends with a sentence about “the width of the new metacarpal relative to dorsopalmar length” referring to observation about “Paleogene sloth material” of Amson et al. (2017). I am puzzled by this sentence, because 1) Amson et al. (2017) referred to a different ratio fo McII, proximodistal length to dorsopalmar depth (character 20 therein); 2) there is no Paleogene taxa studied therein. I actually am not aware of any Paleogene record of sloth Mc II.
6. The MS states that the metacarpal is “missing its distal epiphysis” and that “due to the worn nature of the fossil we cannot rule out the possibility of belonging to a juvenile.” I would like the authors to be more accurate regarding this. Do they consider that the distal epiphysis is unfused and missing (which would suggest an immature ontogenetic stage), or just broken off? I am not entirely sure, but to me the distal end seems ‘crenelated’ and showing the texture of unfused bone surface. So I’d tend to interpret it as an unfused distal epiphysis (hence an immature individual).

Discussion [about the mammalian specimen]

7. The discussion gives a good account of the history of Paleogene specimens that could potentially represent xenarthrans. However, given my assessment of the metapodial affiliation, I cannot agree with the definitive conclusions that are taken here.

Conclusion [about the mammalian specimen]

8. I find the conclusion appropriately more careful about the metapodial attribution. I would recommend to use such a conditional tense for the Discussion as well.

General comments:

9. The mammalian specimen is referred to “Xenarthra” in the abstract, and figure legend, but to a folivoran in the Results, Discussion and Conclusion. That’s misleading, I think. I would recommend to refer to the precise attribution throughout the MS.

Figures

Fig. 1.
• I’d recommend to add a scale and locate North.
• Text refers to two locations/sites. Fig. legend to “the locality”. Are the two sites too close at that scale to be figured separately?
Fig. 2.
• The bone is very nice pictured. I would only recommend adding a label for the trapezoid facet, if that interpretation is kept.
• As stated above, something must be wrong with the specimen’s orientation.

Cited references:

De Iuliis, G., & Cartelle, C. (1994). The medial carpal and metacarpal elements of Eremotherium and Megatherium (Xenarthra: Mammalia). Journal of Vertebrate Paleontology, 13(4), 525–533.
De Iuliis, G., Gaudin, T. J., & Vicars, M. J. (2011). A new genus and species of nothrotheriid sloth (Xenarthra, Tardigrada, Nothrotheriidae) from the Late Miocene (Huayquerian) of Peru. Palaeontology, 54(1), 171–205.
De Iuliis, G., Pujos, F., Toledo, N., Bargo, M. S., & Vizcaíno, S. F. (2014). Eucholoeops Ameghino, 1887 (Xenarthra, Tardigrada, Megalonychidae) from the Santa Cruz Formation, Argentine Patagonia: implications for the systematics of Santacrucian sloths. Geodiversitas, 36(2), 205–255.
Scott, W. B. (1909). Mammalia of the Santa Cruz beds. Part IV. Astrapotheria. Reports of the Princeton University Expeditions to Patagonia., 6, 301–351.


Sincerely,
Eli Amson

Reviewer 2 ·

Basic reporting

References to previous works including fossil reports and stratigraphical context need an update.
The figure of penguins is unnecessary. Materials are not convincing and the authors cannot assign them in any of the cases. The situation is different regarding the metacarpal assigned to Xenarthra, I propose the inclusion of comparative material in this plate.

Experimental design

The Xenarthra is important enough and deserves to be published. However, it receives a deficient treatment.
What this record imply?
The author should compare the metacarpal with other mammals frequent in the Seymour Island levels (e.g. marsupials).

Background is also deficient. Marsupials are not even mentioned.

Validity of the findings

It fails in the backsground and comparisons, see the comments below

Additional comments

Particular comments are below, indicating the line number and the sentence of reference:



58 come from the Upper Eocene La Meseta Formation

The stratigraphy of Seymour Island was carefully studied during years. As a result, a detailed map was made by researchers from Spain and Argentina (Montes et al., 2013).
The Upper Eocene corresponds to Submeseta Formation, see Montes et al. (2013). The authors need to update it.

Montes, M., Nozal, F., Santillana, S., Marenssi, S., & Olivero, E. (2013). Mapa Geológico de la isla Marambio (Seymour) Escala 1: 20.000. Serie Cartográfica Geocientífica Antártica.




64 penguins, including some of the tallest penguins that ever lived

The tallest penguins are described in Acosta Hospitaleche 2014

Acosta Hospitaleche, C. (2014). New giant penguin bones from Antarctica: systematic and paleobiological significance. Comptes Rendus Palevol, 13(7), 555-560.




68 gondwanathere, ungulate, cetacean, and other eutherian mammals (see reviews by Marenssi

You should make a reference of the Dryolestida MLP-91-II-4-3 reported in: MARTINELLI A, CHORNOGUBSKY L, ABELLO A, et al. 2014. The first non-therian dryolestoid from Antarctica. 2014 SCAR Open Science Conference, Auckland, New Zealand, Volume: Abstracts Volume. doi: 10.13140/2.1.2770.8805.




70 Gelfo et al. 2017) as well as non-penguin birds (e.g., Tambussi and Acosta Hospitaleche, 2007;
71 Jadwiszczak et al., 2008; Tambussi and Degrange, 2013; Acosta-Hostpitaleche and Gelfo, 2017)

These references are not the best examples, various relevant contributions are missing, please update.
check also the spelling of: "Hospitaleche and Jadwiszczak along the manuscript"




77 The fossils here described were collected from the Upper Eocene Submeseta Allomember

Now, Submeseta Formation (Montes et al., 2013)



78 (Telm 6/7) of the La Meseta Formation on Seymour Island (Fig. 1).

Take care with this. The authors need to update the stratigraphic scheme. The complete island was mapped, they could assign the place, without hesitations, to Submeseta Allomember II or III.

Fig. 1
This draw is too schematic. As a map, it needs improvement, I dont think it meets the standars of the journal. See comments in the figure.



85 The Submeseta Allomember

again



86 (Priabonian) between 34.96 and 35.13 Ma (Marenssi et al., 1998; Marenssi, 2006).

Many new references! with an update stratigraphic scheme



90 level on the northeastern section of the island (Fig. 1)

I cannot see the locality in this map



95 Geosciences Vertebrate Paleontology Laboratory, Austin, Texas, USA

USA.



102 PILOSA Flower, 1883
103 FOLIVORA Delsuc, Catzefilis, Stanhope, and Douzery, 2001

Please specify the characters that justify this assignment



107 Material – TMM 44190-1, left metacarpal II.

this is an important report, you should make a better comparison (pictures and descriptions)





109 Formation/Age – Submeseta Allomember (Telm 6/7), La Meseta Formation, late Eocene

update!



123 for the metacarpal-carpal complex. TMM 44190-1 is morphologically most consistent with those
124 of Xenarthra (anteaters, armadillos, sloths), especially sloths

The morphology is consistent, therefore it is assigned to a Xenarthra? or it is most consistent?
not the same



140 The majority of mammal fossils from the Eocene of Seymour Island comprise teeth of
141 Astrapotheria (Bond et al., 2011),

This is not true, most of the mammals are marsupials. By the way, considering the importarce of the marsupial record, you should at least mention it. You could also compare the new fossil with marsupials.
See: Goin, F. J., Vieytes C. E., Gelfo, J. N., Chornogubsky, L., Zimicz, A. N. and Reguero, M. A. (2018) New metatherian mammal from the Eocene of Antartida. Journal of Mammalian Evolution

see also Reguero, M., Goin, F., Hospitaleche, C. A., Marenssi, S., & Dutra, T. (2013). West Antarctica: tectonics and paleogeography. In Late Cretaceous/Paleogene West Antarctica terrestrial biota and its intercontinental affinities (pp. 9-17). Springer, Dordrecht.



142 Litopterna (Bond et al., 2006; Gelfo et al., 2015

The authors should also mention the phalanx MLP 13-I-25-2 described in: Gelfo et al., 2015



154 Locality – Seymour Island, Antarctic Peninsula.

This is a complete island, not a locality!



155 Formation/Age – Submeseta Allomember (Telm 6/7), La Meseta Formation, late Eocene.

update



156 Description — TMM 44189-2 preserves

what features support th assignment to Gruiformes?



175 TMM 44189-2 exhibits a combination of character states most similar to those observed
176 in Gruiformes

is this combination unique for these order?



209 Gen. et sp. indet. A

Species from Seymour Island were described from tarsometarsi, why the authors do not assign these materials?



278 represent a juvenile specimen of P. gunnari or a new species within Palaeeudyptes

Juveniles are just a little smaller than adults, the author can evaluate the ontogenetic stage comparing the ossification degree, the end morphology and the textural ageing.



283 Material – TMM 44187-1, partial mandible with associated caudal fragments

fragments of what?



284 Locality – Seymour Island

locality!




300 195, IB/P/B-0653; Jadwiszczak 2006; Acosta Hospitaleche and Haidr, 2011)

update! papers including comparable materials were posteriorly published



316 preservation, TMM 44187-1 is not considered referable to any known Eocene taxa at this time.

A single Eocene species (Anthropornis grandis) from Seympur Island has cranium, mandible and postcranial associated elements (Acosta Hospitaleche et al., 2019). Therefore, mandibles can only be assigned to A.grandis, all the other known mandibles are isolated.



324 penecontemporaneous fossil discoveries from Patagonia (see Reguero et al., 2002; 2014).

avian references from Chile and Argentina are missing, where the same Antarctic species were reported

see:

Yury-Yáñez, R. E., Otero, R. A., Soto-Acuña, S., Suárez, M. E., Rubilar-Rogers, D., & Sallaberry, M. (2012). First bird remains from the Eocene of Algarrobo, central Chile. Andean Geology, 39(3), 548-557.

Acosta Hospitaleche, C., & Olivero, E. (2016). Re-evaluation of the fossil penguin Palaeeudyptes gunnari from the Eocene Leticia Formation, Argentina: additional material, systematics and palaeobiology. Alcheringa: An Australasian Journal of Palaeontology, 40(3), 373-382.



331 vertebra and tooth consistent with referral to a basilosaurid archaeocete and a previously
332 described ungulate species, respectively.

Please provide a picture, a more precise systematic reference and the material number. On the contrary, reader cannot identify these records.



360 (Tambussi and Acosta Hospitaleche, 2007),

update references! a decade of research and analysis is ignored here.




384 The penguin mandible described here is the largest and most complete from a spear-billed
385 penguin yet reported from the Eocene of Antarctica.

not true, see Acosta Hospitaleche et al., 2018 (abstract) and Acosta Hospitaleche et al., 2019 (paper)




399 and larger than other Eocene Antarctic spear-bills recovered (Acosta Hospitaleche and
400 Jadwiszczak, 2011).

see also Acosta Hospitaleche et al., 2019



404 from Seymour Island alone (Jadwiszczak, 2006; Ksepka and Clarke, 2010; Acosta Hospitaleche,
405 2013 and references therein).

it needs update. Acosta Hospitaleche et al., 2017 describe a new species, and other contributions of Jadwiszczak discuss the known diversity reporting new material that could belong to new species.



418 during the Eocene and support previously controversial reports of Gruiformes and Xenarthra. A
419 metacarpal is proposed to possibly represent the first record of Folivora and lends support to
420 previously reported xenarthran materials that have been subsequently questioned or lost.

It is assigned without doubts in the Systematic Paleontology



FIG1: The map needs some work. Please include coordinates, scale, reference points, north.
FIG.2: A comparative plate including other Pilosa and Vermilingua will be useful (something similar to Fig 3)
Fig. 4: Some materials are too fragmentary (e.g. E-F), considering that thousands of penguin remains are known from these units, it is not worth photographing this material.

---

## Round 0.2 · Minor Revisions

Thank you for your detailed attention to the comments from the reviewers on the first version of the manuscript. After a second round of review (one of the original two reviewers participated), only a handful of very minor revisions are now needed. These are largely issues of wording, spelling, etc., and I hope will not require a significant amount of time. Once I have your revision, I should be able to return a final decision in very short order.

One final, completely optional, item to consider is deposit of the 3D data. The supplemental 3D PDF is a great feature of the current manuscript; if permitted by institutional policy, I suggest also depositing the original surface data (i.e., in STL or OBJ format) at a venue such as MorphoSource or at your institutional archive. This is, again, not required for acceptance, but would be a helpful service for the community if you have the ability to do so (and the data can be restricted for download/printing/etc. if required by institutional policy).

·

Basic reporting

no comment

Experimental design

no comment

Validity of the findings

no comment

Additional comments

Dear Editor, dear Authors,

I reviewed the revised manuscript titled New mammalian and avian records from the late Eocene La Meseta Formation of Seymour Island, Antarctica. The authors have carefully addressed all the concerns raised at the first round of reviews, so I strongly encourage publication of the manuscript, after consideration of minor issues I raise below:

The mammal specimen treatment was improved thanks to a more thorough comparison and a less decisive attribution. I might just suggest, given the revised comparison, to refer the specimen to ?XENARTHRA, to convey this more careful attribution.

228: typo in “Glyptotherium”

The reference referred to as Hatcher et al. (1903) (l.251 for instance) should be Scott (1903-1905)

250 “the facet also has a ridge located along the dorsal surface that is not seen in the new fossil”
The ridge that I think the authors are referring to is a synostosis scar, of variable development (individually), so I would not consider it as an important character to mention.

265 but closer matches
“but matches more closely”?

268: typo in “De Iuliis”

723: italicize “Anthropornis Grandis”

855: italicize “Neoglyptatelus”

859: typo for “Miocene”

912: Harcher: see above

969: “Pascual, R..” (double period)

1028: “Woodburne, MO..,” (double period)

References to doi not consistent (e.g., 826 and 883) and given just for a few refs.

Figure 2 caption: “mcc” could be replaced by “?mcc”, and reference to the fact that it could be the Mc I could be added there as well. Moreover, I’m not sure why the trapezoid facet is abbreviated including the word “facet” (which is not the case for the other facets).

Sincerely,
Eli Amson

---

## Round 0.3 · accepted · Accept

Thank you for your careful consideration and incorporation of the most recent round of comments. You have done an excellent job of responding to the reviewers.